# SuperPure: Efficient Purification of Localized and Distributed Adversarial Patches

## Abstract

As vision-based machine learning models are increasingly integrated into autonomous systems, concerns about physical adversarial patch attacks are growing. While state-of-the-art defenses can achieve certified robustness with minimal impact on utility against highly concentrated localized patch attacks, they fall short in two important areas: *(i)* they are vulnerable to low-noise *distributed* patches, where perturbations are subtly dispersed to evade detection or masking, as shown recently by the DorPatch (He et al., 2024) attack; *(ii)* they are extremely time- and resource-consuming, making them impractical for latency-sensitive applications.

To address these challenges, we propose *SuperPure*, a defense that combines pixel-wise adversarial masking with GAN-based super-resolution. Specifically, we measure SuperPure's robustness against three attack families—*distributed*, *localized*, and *natural* patches—and verify its plug-and-play effectiveness on three distinct backbones (ResNet, ViT, and EfficientNet) across multiple patch sizes. Extensive ImageNet evaluations show that SuperPure: *(i)* boosts robustness against localized patches by $>20\%$ and clean accuracy by $\sim10\%$; *(ii)* achieves 59% robustness against distributed patches (vs. 0% for PatchCleanser) and 43.2% against natural patches while preserving clean utility; *(iii)* runs over $10\times$ faster than state-of-the-art defenses on a Jetson Orin Nano. Code will be released upon acceptance.

## 1 Introduction

Deep learning models have achieved remarkable success in various computer vision tasks, including image classification, object detection, and semantic segmentation (Krizhevsky et al., 2012; He et al., 2016a; Dosovitskiy et al., 2021). However, they are highly vulnerable to adversarial attacks, where carefully crafted perturbations are added to inputs to cause misclassification (Szegedy et al., 2014; Goodfellow et al., 2015; Madry et al., 2018). While many attacks involve imperceptible $\ell_p$-bounded perturbations spread across an image (Croce & Hein, 2020), adversarial patches instead introduce large, visible perturbations confined to a region of the image (Hayes, 2018; Karmon et al., 2018). These pose serious threats in the physical world: attackers can print and place a patch on an object to manipulate a model's prediction, as demonstrated on road signs and objects in autonomous driving scenarios (Brown et al., 2017; Wei et al., 2022; Eykholt et al., 2018; Liu et al., 2018). Beyond simple patches, black-box methods such as PatchAttack use reinforcement learning to generate effective patches without gradient access (Yang et al., 2020), while TnT shows that patches can be made naturalistic, resembling benign objects like flowers, yet still remain adversarial (Doan et al., 2022). Furthermore, distributed attacks such as RP2 (Eykholt et al., 2018) and DorPatch (He et al., 2024) spread perturbations across multiple regions, making them harder to detect or cover, and enabling occlusion-robust real-world deployment.

Various defense strategies have been proposed to counter adversarial patches (Jing et al., 2024; Chiang et al., 2020; Xiang et al., 2021; 2022; 2024; Mao et al., 2022). Early approaches include Digital Watermarking (Hayes, 2018) and Local Gradient Smoothing (Naseer et al., 2019), which attempt to suppress suspicious regions; however, adaptive attacks can bypass these methods (Chiang et al., 2020). Certified defenses such as IBP extensions (Chiang et al., 2020), Clipped BagNet (Zhang et al., 2020), and PatchGuard (Xiang et al., 2021) leverage bounded receptive fields but require architectural changes. PatchCleanser (Xiang et al., 2022) advanced this line of research by applying

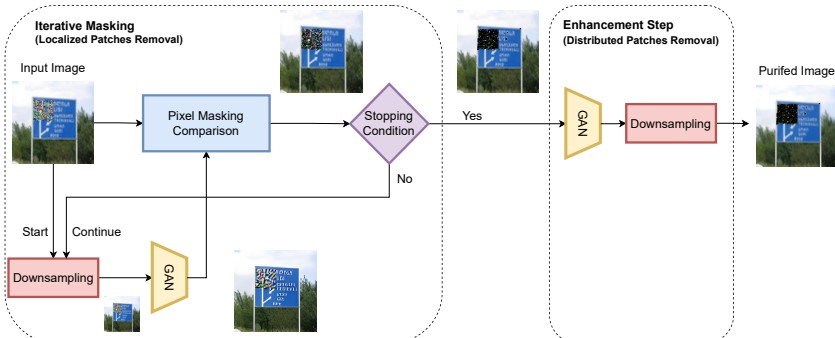

Figure 1: SuperPure pipeline: at each iteration, we downsample, GAN-upsample, and mask high-residual pixels. If newly masked pixels exceed the threshold, we repeat. At the end, a final upsample–downsample "enhancement" removes subtle perturbations.

randomized masking with formal guarantees, and PatchCure (Xiang et al., 2024) optimized it for efficiency, albeit at the cost of retraining. Nonetheless, both fail against disturbed distributed patch attacks like DorPatch (He et al., 2024). Patch localization defenses, including SentiNet (Chou et al., 2020), Jedi (Tarchoun et al., 2023), PatchZero (Xu et al., 2023), and PAD (Jing et al., 2024), attempt to detect and remove malicious regions, but often depend on assumptions about patch appearance or location. Adversarial training has also been explored for patches (Rao et al., 2020; Wu et al., 2020; Addepalli et al., 2021), but is computationally costly.

Another line of work leverages image super-resolution (SR) and image purification as a defense strategy, where SR has been used to project adversarial examples back onto the manifold of natural images. Mustafa et al. (2019) showed that SR can mitigate subtle $\ell_p$ perturbations, and DiffPure (Nie et al., 2022b) used diffusion models to remove dispersed noise. However, both SR- and diffusion-based defenses often fail on high-magnitude localized patches and sometimes sharpen the adversarial signal instead (Nie et al., 2022b; Mao et al., 2022). This leaves patch attacks—especially distributed ones—largely unresolved.

Apart from robustness, computation overhead is another critical limitation. Defenses like Patch-Cleanser, PatchCure, and DiffPure can take tens of seconds per image (Xiang et al., 2022; 2024), which is impractical for latency-sensitive applications such as autonomous driving.

**Research Gap.** Despite notable progress, a unified and practical defense for physical patch attacks remains absent—one that is robust to ***both*** localized and distributed patches while incurring minimal latency and resource overhead.

**Our Solution.** We propose *SuperPure*, an iterative, plug-and-play defense that couples downsampling with GAN-based SR and residual-driven pixel masking. The key idea is to exploit the natural-image prior learned by the SR generator: given a patched input $x$, the reconstruction $z = G(D(x))$ (i.e., SR $G$ after downsampling $D$) tends to modify patch-corrupted pixels far more than clean pixels. We compute a per-pixel residual $r = |z - x|$ and build a mask $m = 1[r > \tau]$ (e.g., a threshold). High-residual pixels—likely influenced by the patch—are overwritten by the SR reconstruction, updating $x \leftarrow m \odot z + (1 - m) \odot x$; low-residual pixels remain unchanged. Iterating this procedure progressively removes adversarial signal while preserving benign content, yielding an efficient and architecture-agnostic defense. This process is shown in Figure 1.

We provide extensive results and analysis for various patches and defenses. Compared to state-of-the-art methods such as PatchCleanser (Xiang et al., 2022) and PAD (Jing et al., 2024), *SuperPure* improves robustness against localized patches by over 20%, attains 58% robust accuracy under distributed attacks (where PatchCleanser fails), boosts clean accuracy by nearly 10%, and reduces end-to-end latency by more than 98%.

## 2 PROBLEM FORMULATION

**Attack Objectives.** The attacker aims to mislead a classifier $\mathcal{F}$ by embedding adversarial patches into clean images. Given an input $\mathbf{x} \in \mathbb{R}^{H \times W \times C}$ with label $y$, an adversarial image is constructed as $\mathbf{x}_{\text{adv}} = \mathbf{x} + \boldsymbol{\delta}$, where $\boldsymbol{\delta}$ denotes the patch perturbation. The objective is to ensure $\mathcal{F}(\mathbf{x}_{\text{adv}}) \neq y$. We

consider two cases: *localized patches*, confined to a contiguous region $\mathcal{P}_L$, and *distributed patches*, dispersed across multiple regions $\mathcal{P}_D = \{p_i\}_{i=1}^n$, as in DorPatch (He et al., 2024), which spreads perturbations to evade detection. We assume *white-box* access to the classifier's gradients and, in adaptive settings, knowledge of *SuperPure*'s parameters.

**Defense Objectives.** The defender seeks to remove both localized and distributed patches while preserving accuracy on clean data. Given a clean input $(\mathbf{x}, y)$, the defense should maintain $\mathcal{F}(SuperPure(\mathbf{x})) = y$, while for adversarial inputs $\mathbf{x}_{adv}$ it should recover correct classification, i.e., $\mathcal{F}(SuperPure(\mathbf{x}_{adv})) = y$.

**System Objectives.** For practical deployment, *SuperPure* is (i) *model-agnostic*—requiring neither retraining nor architectural modifications—and (ii) *compute-efficient*, supporting high-resolution inputs with real-time throughput on resource-constrained hardware (e.g., mobile-class GPUs such as the NVIDIA Jetson Nano family).

## 3 METHOD

In this section, we propose *SuperPure*, a defense strategy against adversarial patch attacks that combines a downsampling and upsampling process with a pixel-by-pixel comparison to remove both singular and distributed adversarial patches. The details of our method are presented in Algorithm 1 (presented in the Appendix) and illustrated in Figure 1.

Briefly, our algorithm first downsamples an image, $x_{adv}$, by a factor of $s$ (=4 in our setup). The downsampled image, $x_{down}$ is then fed into an upsampling method to generate a new image, $x_{up}$. The upsampled image has the same size as the original (i.e., $|x_{up}| = |x_{adv}|$). While various upsampling/image generation mechanisms exist, our experiments reveal that a GAN-based super-resolution balances accuracy, robustness, and latency. It further introduces non-linearity that helps remove the patches while retaining the information about the original image. The new image, $x_{up}$, is compared, pixel-wise, with $x_{adv}$ using a threshold, $\lambda$. This process (downsampling, upsampling, and masking) repeats multiple times until the number of masked pixels becomes smaller than a threshold, $\epsilon$. The final step involves first upsampling and then downsampling the image. No masking is applied during this step. The purpose of this final step is to *(a)* enhance the image quality for improved accuracy and *(b)* eliminate low-noise distributed patches.

We explain the details of the downsampling process in Section 3.1 and then discuss the details of the super-resolution and masking algorithm in Section 3.2. In Section 3.3, we outline how the method is iterated and specify the stopping condition for these iterations, and in Section 3.4, we describe the enhancement step. Further optimizations and detailed analysis are described in the Appendix.

### 3.1 DOWNSAMPLING STEP

Downsampling an image reduces its spatial resolution, leading to the loss of high-frequency information; this disproportionately affects adversarial patches in comparison to the essential image content as explained by Xu et al. (2018) and Shannon (1949). This is due to adversarial patches being heavily reliant on precise, high-frequency perturbations (Brown et al., 2017). For downsampling, we utilize *bilinear interpolation*, which estimates each output pixel as a weighted average of the surrounding input pixels (Youssef, 1998). This averaging process smooths out the detailed perturbations in the patch, causing the high-frequency information needed for the patch's effectiveness to be lost. In contrast, natural images typically display redundancy and correlation among neighboring pixels (Wang et al., 2004; Torralba & Oliva, 2003). Important features and structures are often replicated across the image, making essential information less susceptible to significant degradation.

Formally, for an image of size $H \times W$, downsampling can be modeled as a function $D_s : \mathbb{R}^{H \times W \times C} \to \mathbb{R}^{h \times w \times C}$, where $s$ is the scaling factor ($s > 1$), $h = H/s$, and $w = W/s$. Applying downsampling to the adversarial image yields:

$$\mathbf{x}'_{adv} = D_s(\mathbf{x}_{adv}) = D_s(\mathbf{x} + \boldsymbol{\delta}) = D_s(\mathbf{x}) + D_s(\boldsymbol{\delta}). \tag{1}$$

Then, the energy of the adversarial patch after downsampling can be approximated as:

$$\|D_s(\boldsymbol{\delta})\|_2^2 \approx \frac{1}{s^2} \|\boldsymbol{\delta}\|_2^2, \tag{2}$$

indicating a substantial decrease in the perturbation's magnitude due to spatial averaging.

This theoretical insight aligns with empirical evidence. Pixels within adversarial patches consistently exhibit significantly higher reconstruction errors compared to non-patch pixels; in our experiments, patch regions showed approximately $8\times$ ***higher errors*** (mean squared error of 0.6054 vs. 0.0829, as shown in Figure 7 in Section A.4). Additionally, prior work by Guo et al. (2018) similarly demonstrated that downsampling (e.g., via JPEG compression) effectively reduces adversarial perturbation effectiveness while preserving high accuracy on clean images. Nonetheless, simple downsampling alone remains vulnerable to adaptive attacks, underscoring the necessity of our proposed iterative and non-linear GAN-based upsampling process.

## 3.2 UPSAMPLING AND MASKING

While downsampling degrades adversarial patch regions more significantly than natural non-patch areas, our initial analysis showed that applying naive downsampling and upsampling is not sufficient to fully remove distributed patches and stronger adaptive attacks. Instead, a more powerful transformation is needed. The key idea is to apply a transformation that ***disproportionally*** *affects adversarial regions over benign regions*.

Based on this insight, we propose utilizing a super-resolution model for upsampling. While there are various super-resolution models available, we have chosen Real-ESRGAN (Wang et al., 2021) as our primary model. Real-ESRGAN is a GAN-based model designed and trained to reconstruct high-quality images from low-resolution inputs; we use a pre-trained model provided by the authors, with no further fine-tuning on our datasets.

There are two reasons why the GAN struggles to reconstruct patch regions in comparison to benign non-patch regions. *First*, for a pixel $p_{i,j}$ within the adversarial patch, the surrounding pixels $p \in B((i,j),\tau)$ are typically uncorrelated with $p_{i,j}$ and are also heavily degraded, leaving the GAN $G$ with insufficient information for precise reconstruction. On the other hand, natural non-patch regions have more structural coherence and redundancy, leaving enough information for reconstruction even after downsampling. *Second*, since the GAN is trained on a dataset of natural images, it is optimized to generate outputs that align with the natural image distribution. However, adversarial patches deviate from this distribution, resulting in a greater pixel variation in these regions in comparison to the rest of the image. Let $p_{i,j_a}$ denote an adversarial pixel and $p_{i,j_c}$ a clean pixel inside an image $\pi$. This relationship can be expressed through the following inequality:

$$\mathbb{E}(|p_{i,j_a} - G(D_s(\pi))_{i,j}|) \gg \mathbb{E}(|p_{i,j_c} - G(D_s(\pi))_{i,j}|). \tag{3}$$

This disparity results in *greater pixel-wise differences* between the original image and the reconstructed output in patch regions compared to non-patch regions. Therefore, by comparing the original adversarial image $\mathbf{x}_{\text{adv}}$ to the upsampled image $\mathbf{x}_{\text{up}}$ on a pixel-by-pixel basis, we can mask adversarial regions by identifying where the reconstruction error exceeds a pre-defined threshold $\lambda$. Specifically, a pixel-wise comparison computes the $\mathcal{L}_2$-distance between corresponding pixels in $\mathbf{x}_{\text{adv}}$ and $\mathbf{x}_{\text{up}}$. A binary mask $\mathbf{m}$ is then generated, where:

$$\mathbf{m}(i,j) = \begin{cases} 1, & \text{if } \|\mathbf{x}_{\text{adv}}(i,j) - \mathbf{x}_{\text{up}}(i,j)\|_2 > \lambda, \\ 0, & \text{otherwise.} \end{cases}$$

The binary mask $\mathbf{m}$ is then overlaid on the adversarial image $\mathbf{x}_{\text{adv}}$, effectively masking the regions that exhibit high reconstruction error and are likely to be adversarial patches.

## 3.3 ITERATION AND STOPPING CONDITION

The inequality from Equation (3) suggests that with an appropriate choice of $\lambda$ we can effectively mask adversarial pixels while preserving clean ones. However, a single iteration may not suffice to identify most adversarial pixels beyond the threshold, as a GAN can utilize local context. As a result, we use multiple iterations and observe that as the number of iterations increases, our model converges and progressively masks out adversarial pixels. The progressive masking can be seen in Figure 2. This is also demonstrated empirically in Figure 3 as we can see the total masked pixels and total new pixels converge and steady as the number of iterations increases.

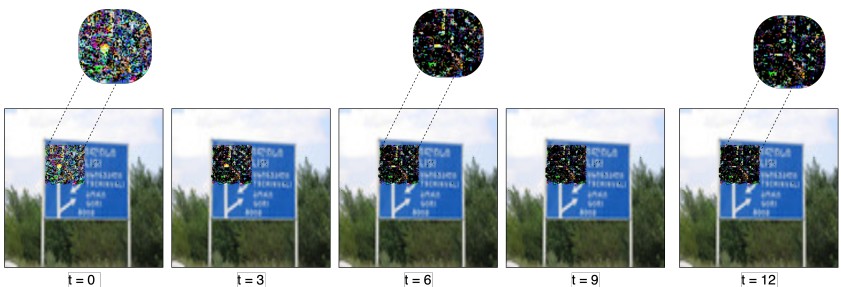

Figure 2: The masking process of SuperPure across multiple time steps.

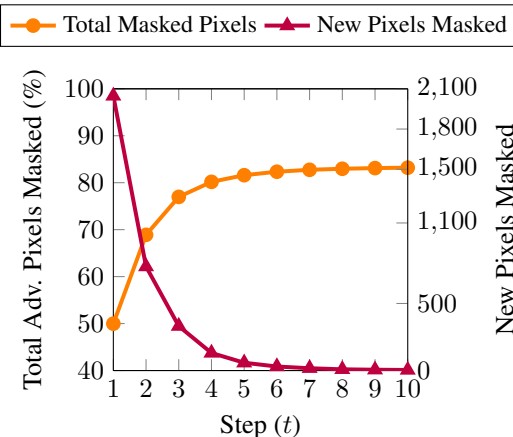

Figure 3: Total and new adversarial pixels masked (ImageNet, Patch Size 64, Threshold=0.7).

Based on the insight above, the process is repeated to ensure that the patch is as completely masked as possible. At each iteration, the adversarial image $\mathbf{x}_{\text{adv}}$ is updated by overlaying the binary mask $\mathbf{m}$, and the new masked image becomes the input for the next iteration. This allows the adversarial regions to be gradually refined and more effectively suppressed with each cycle. As presented in Figure 2, with each iteration, a greater percentage of the patch regions in the image is masked. The majority of the masking is done in earlier iterations; the number of newly masked pixels steadily decreases until the stopping condition is met.

The stopping condition is based on the fraction of pixels that have been newly masked at a given iteration. Specifically, the iteration continues until the percentage of the newly masked pixels at iteration $t$ is below a threshold $\epsilon$:

$\frac{\sum_{i,j}^{W,H} \mathbf{m}_{\mathbf{t}}(i,j)}{W \times H} < \epsilon$, where $W$ and $H$ are the width and height of the image. Once the stopping condition is met, perceptible adversarial patches are mostly eliminated.

## 3.4 Removing Small Distributed Perturbations

While the aforementioned iterative masking process is effective at removing perceptible adversarial patches, it struggles with smaller, distributed perturbations, such as those presented in Dorpatch (He et al., 2024). These perturbations are subtler and often resemble natural variations within the image. Because our process relies on the assumption that the reconstruction of adversarial patches will differ significantly from that of non-patch natural image regions, this limitation of the masking process is expected.

A simple solution to mitigating these attacks involves upsampling the image via a GAN-based super-resolution model and then downsampling to the original size. This upsampling step via super-resolution generates additional high-frequency details, while the subsequent downsampling removes that added information. The approach can be compared to DiffPure (Nie et al., 2022a), where random noise is added to an image before denoising it to remove any adversarial noise process. In our case, upsampling acts as the *diffusion* step, and downsampling serves as the denoising step, eliminating the small adversarial perturbations that are embedded within the image. Using Equation (3) and its assumptions, we observe that the generated pixel is more likely to resemble the natural distribution, excluding adversarial noise. The averaging process during downsampling further reduces this noise by smoothing out pixel-level variations, while the new clean pixel generated by the GAN helps to replace adversarial components. Together, these steps effectively reduce the overall noise, acting as a surrogate for the denoising step in diffusion models.

This process is performed after the initial iterative masking step, ensuring that any remaining subtle adversarial perturbations are addressed. We chose to upsample the image before downsampling (instead of vice versa) because starting with downsampling discards significant information in the image, leading to a loss of essential details and poor classifier performance. The final processed image can then be used for downstream tasks, ensuring that the adversarial manipulations no longer have a significant effect on model performance.

# 4 RESULTS

In this section, we provide a comprehensive evaluation of *SuperPure*. We describe the experimental setup in Section 4.1, followed by a presentation of the robustness analysis against various patch attacks in Section 4.2. Additional sensitivity and ablation studies are also provided in the Appendix.

## 4.1 EXPERIMENTAL SETUP

We evaluate *SuperPure* against adversarial patch attacks using a subset of ImageNet (Deng et al., 2009), where five validation images per class (5,000 total) are selected for diversity.

To demonstrate classifier-agnostic robustness, we test on three distinct architectures: EfficientNet-B0 (Tan & Le, 2019), ResNet-152 v2 (He et al., 2016b), and ViT-B/16 (Dosovitskiy et al., 2021), each with publicly available PyTorch ImageNet checkpoints (Paszke et al., 2019). These models span compound scaling (EfficientNet), deep residual learning (ResNet), and patch-based Transformers (ViT), providing broad coverage of design paradigms.

We use Real-ESRGAN (Wang et al., 2021) for super-resolution, chosen for its efficiency, lack of ImageNet training overlap, and reproducible checkpoints. We use ×2 and ×4 upsampling models trained on DIV2K, Flickr2K, and OST (Agustsson & Timofte, 2017; Timofte et al., 2017; Wang et al., 2018), noting ImageNet was excluded. These scales provide the best robustness–latency tradeoff in preliminary analysis. Unless otherwise stated, we set $\lambda = 0.7$ and $\epsilon = 4$ (pixels).

### 4.1.1 ADVERSARIAL ATTACKS

To evaluate our model's performance, we subject it to *three distinguished attack families*: (i) **localized patch attacks**, (ii) **distributed patch attacks**, and (iii) **natural patch attacks**. The concrete instantiations used for each family are detailed below.

For *localized* patches, we follow PatchCleanser (Xiang et al., 2022) and PatchGuard (Xiang et al., 2021), and generate adversarial patches using the Masked PGD method.

For *distributed* patch attacks, we apply the Dor-Patch attack (He et al., 2024), which uses a patch budget of 12%, a perturbation density of 0.1%, and up to 5,000 optimization iterations.

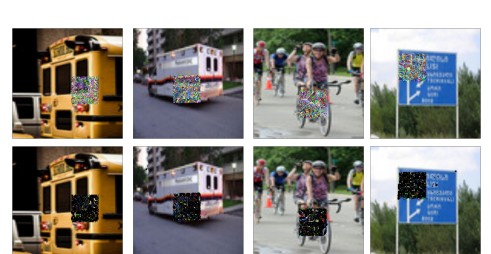

For *naturalistic* adversarial patches, we adopt the TnT Attacks method proposed by Doan et al. (2022). TnT generates a single, visually innocuous patch (e.g., a flower) using a GAN, which reliably misleads classifiers when placed at arbitrary locations. Following the official implementation, we optimize one targeted patch for 5,000 SGD steps (step size 0.05) on 5,000 selected Im-

Figure 4: Quality results for *SuperPure* before (top) and after purification (down).

ageNet images. The final patch, approximately 96×96 pixels (∼14% of the input).

### 4.1.2 EVALUATION METRICS AND HARDWARE SETUP

We evaluate our defense using clean accuracy (top-1), robust accuracy, and per-example inference time (s/img) to assess both effectiveness and computational overhead. Clean accuracy is the proportion of clean test images correctly classified, while robust accuracy measures accuracy under adversarial attacks. For comparison, we also report results of three state-of-the-art defenses—PAD (Jing

Table 1: Robustness of different defense methods against localized patch attacks, distributed patch attacks, and natural patch attacks using a ResNet-152 backbone. Latency (Lat) is measured on Jetson Orin Nano.

| Defense Method | Clean Acc ↑ | Robustness ↑ | | | Lat (s) ↓ |
|---|---|---|---|---|---|
| | | Localized | Distributed | Natural | |
| No Defense | 71.7% | 24.5% | 0% | 15.4% | 0.10 |
| PatchCleanser (Xiang et al., 2022) | 68.9% | 51.5% | 0% | 39.8% | >50 |
| PAD (Jing et al., 2024) | 48.2% | 50.3% | 39% | **48%** | 8.8 |
| PatchCURE (Xiang et al., 2024) | 72% | 41% | 0% | 37% | 12 |
| *SuperPure* (ours) | **79.8%** | **76.3%** | **59%** | 43.2% | **0.72** |

et al., 2024), PatchCleanser (Xiang et al., 2022), and PatchCURE (Xiang et al., 2024)—using their optimal settings as described in their respective papers. Since PatchCleanser is sensitive to patch size, we tested multiple configurations and report the best-performing result; PatchCURE improves runtime and model performance by modifying the backbone (e.g., ViT-SRF) while maintaining robustness. Figure 4 illustrates qualitative purified examples. All evaluations are conducted on an NVIDIA RTX 4090 GPU (24 GB, PyTorch 1.12), and latency is measured on an NVIDIA Jetson Orin Nano to reflect realistic edge-device performance.

## 4.2 ROBUSTNESS RESULTS

### 4.2.1 OVERALL ROBUSTNESS AND EFFICIENCY COMPARISON

To provide a clear comparison with state-of-the-art methods, we evaluate all defenses against three distinct patch attack categories: *localized adversarial patches* (Masked-PGD, $32 \times 32$), *distributed adversarial patches*, DorPatch (He et al., 2024), 12% budget, 0.1% density), and *naturalistic adversarial patches*, TnT (Doan et al., 2022), natural-looking flower patch, approximately 14% area). Table 1 summarizes results for ResNet-152 on 5,000 ImageNet validation images, including clean accuracy, robustness, latency measured on a Jetson Orin Nano.

Overall, results show that *SuperPure* remains consistently accurate against different patches as *opposed* to prior work. With a single $32 \times 32$ Masked-PGD patch, *SuperPure* maintains 76.3% robust accuracy, more than triple the undefended model and *25% higher* than PatchCleanser.

Crucially, our approach offers a substantially ***more robust*** model against *distributed* attacks, while certified robustness methods (PatchCleanser and PatchCure) are at 0%, and PAD is only 39%. *SuperPure*, however, still achieves 59% robust accuracy. Our enhancement step detects and denoises the low-amplitude perturbations caused by DorPatch, enabling the masking stage to eliminate any remaining artifacts effectively.

For natural patches, PAD has the highest robustness (48%) while *SuperPure* has a relatively similar robustness (43.2%). Compared to PAD, *SuperPure* is more than **10x faster** and is about **30% more accurate** on clean accuracy, making it a significantly better choice in real-world environments where latency-accuracy-robustness needs to be jointly optimized. (See Figure 5 for an example of *SuperPure* removing a natural patch.)

Latency underscores the practicality of our approach. On a Jetson Orin Nano, PatchCleanser can spend close to a minute per frame, and PatchCURE twelve seconds, whereas *SuperPure* completes purification in just 0.72s. Crucially, *SuperPure* retains its plug-and-play nature; PatchCURE trims runtime only by retraining the classifier, a major limitation in many deployments.

Finally, it is worth underscoring how *SuperPure* behaves on *benign* inputs. Instead of negatively impacting accuracy, our GAN-based enhance-

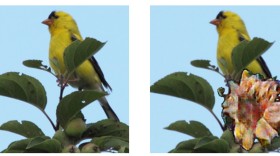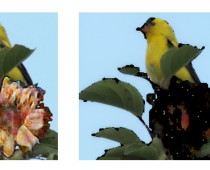

(a)  (b)  (c)

Figure 5: Qualitative result for naturalistic (TnT) patches. (a) Clean image; (b) image with TnT flower patch; (c) Output of our *SuperPure*, which suppresses almost all patch pixels while preserving fine details in the benign regions.

Table 2: Robustness of different defense methods against single (localized) adversarial patch attacks on ViT, EfficientNet, and ResNet classifiers.

| Model | Defense Method | Patch Size | | | | | |
|---|---|---|---|---|---|---|---|
| | | 0 (no attack) | 16×16 | 32×32 | 48×48 | 64×64 | 96×96 |
| ViT | No Defense | 74.84 | 38.02 | 4.32 | 00.50 | 0.16 | 0.00 |
| | PatchCleanser (Xiang et al., 2022) | 72.10 | 54.33 | 44.21 | 35.3 | 30.74 | 20.72 |
| | PAD (Jing et al., 2024) | 44.76 | 46.58 | 47.04 | 46.36 | 45.62 | 41.64 |
| | *SuperPure* | **82.98** | **80.70** | **77.82** | **77.52** | **74.66** | **65.9** |
| EfficientNet | No Defense | 60.76 | 30.82 | 5.12 | 0.82 | 0.20 | 0.02 |
| | PatchCleanser (Xiang et al., 2022) | 57.98 | 43.60 | 38.46 | 27.46 | 22.66 | 10.92 |
| | PAD (Jing et al., 2024) | 34.70 | 35.30 | 34.70 | 33.94 | 32.72 | 26.02 |
| | *SuperPure* | **69.08** | **63.54** | **60.48** | **54.12** | **46.72** | **28.22** |
| ResNet | No Defense | 71.70 | 45.10 | 24.52 | 14.28 | 4.38 | 0.10 |
| | PatchCleanser (Xiang et al., 2022) | 68.98 | 56.72 | 51.64 | 41.28 | 33.82 | 19.66 |
| | PAD (Jing et al., 2024) | 48.19 | 50.10 | 50.28 | 49.38 | 45.04 | 43.30 |
| | *SuperPure* | **79.86** | **76.74** | **76.30** | **74.20** | **70.64** | **57.84** |

ment *improves* the ResNet-152 baseline from 71.7% to 79.8%. PatchCleanser loses about two percentage points, whereas PAD reduces the accuracy by more than 20%.

### 4.2.2 ROBUSTNESS ACROSS PATCH SIZES AND BACKBONES

Table 2 reports clean and robust Top-1 accuracy for ViT-B/16, EfficientNet-B0, and ResNet-152 across six patch sizes—from 16×16 to 96×96. In every setting, *SuperPure* outperforms prior defenses, i.e., PatchCleanser (Xiang et al., 2022) and PAD (Jing et al., 2024), by a wide margin.

**Small patches** (16×16). *SuperPure* retains 80.7% accuracy on ViT, 63.5% on EfficientNet, and 76.7% on ResNet—more than doubling the no-defense baseline and exceeding PAD by >25% on all three backbones.

**Medium patches** (48×48). At this more challenging size, *SuperPure* still delivers 77.5% (ViT), 55.0% (EffNet), and 74.2% (ResNet), while the strongest baseline (PAD) tops out at 49.4% on ResNet and 33.9% on EfficientNet.

**Large patches** (96×96). Even when nearly one-fifth of the image is corrupted, *SuperPure* maintains 65.9% (ViT), 58.2% (ResNet), and 28.2% (EffNet) accuracy—contrasting sharply with Patch-Cleanser's <20% and PAD's 41.6% (ViT) and 43.3% (ResNet).

**Clean-image performance.** On benign inputs (*patch size=0*), *SuperPure improves* accuracy relative to the undefended classifier by roughly 8% on every backbone (e.g., 83% on ViT versus 74.8% baseline). Hence, the super-resolution enhancement stage (Section 3.4) yields a dual benefit: higher benign accuracy and markedly stronger robustness. We quantify this trade-off further in the ablation study (Section A.7).

### 4.2.3 ADAPTIVE WHITE-BOX ATTACKS

We also evaluate our defense under an adaptive *white-box* threat model, where the adversary has complete access to both our purification network (including the super-resolution module and our method) and the target classifier. This means the attacker can compute gradients through every component of our defense to craft adversarial patches tailored explicitly to our defense mechanism.

Table 3: Robust Accuracy (%) under White-Box Attacks on ResNet.

| Defense | Patch Size | |
|---|---|---|
| | 48×48 | 64×64 |
| Naïve Down&Up (white-box) | 9.12 | 4.89 |
| PatchCleanser (non–white-box) | 41.28 | 31.28 |
| *SuperPure* (ours, white-box) | **60.38** | **51.52** |

As shown in Table 3, which compares the robust accuracy of three different defenses—*Naïve Down&Up*, *PatchCleanser*, and our method *SuperPure*—against adversarial patches of size 48 × 48

and $64 \times 64$ on ResNet[1], a simple "Naïve Down&Up" defense completely fails when facing an adaptive attacker. This naive approach applies a fixed downsampling followed by upsampling in hopes of smoothing out adversarial noise. However, under a white-box setting where the attacker has the knowledge of this defense, it becomes trivial for the attacker to generate perturbations that survive such transformations. As a result, the robust accuracy drops sharply to only 9% and 5% for patch sizes of $48 \times 48$ and $64 \times 64$, respectively.

PatchCleanser, which is not differentiable and thus not directly applicable in a white-box setting, is evaluated here under its own original (black-box) setup. While it performs better than the naive method in that setting (41.28% and 31.28%), it still falls short compared to our method. In contrast, our proposed *SuperPure* is evaluated in a **_fully adaptive white-box setting_**, where the attacker has complete access to the super-resolution model and can backpropagate through the entire pipeline. Despite this, our method maintains robust accuracy of 60.38% and 51.52%, significantly outperforming both the naive baseline and PatchCleanser—even though the latter operates under a more favorable (black-box) scenario. This strong robustness, even under white-box conditions, stems from the inherent nonlinearity and complexity of our defense pipeline. Unlike simple smoothing operations, our method first projects inputs into a more natural image manifold using a deep super-resolution network, then applies a masking and enhancement mechanism that further disrupts adversarial structures. Because the entire purification process is nonlinear and includes operations that do not preserve gradients in a predictable way, it becomes significantly harder for the attacker to generate perturbations that survive all these transformations. As a result, even with full access to our model and the ability to compute gradients through it, the adversary struggles to construct successful attacks. The resulting drop in robust accuracy compared to the black-box setting is relatively small, demonstrating that our method retains strong practical resilience—even in challenging adaptive white-box scenarios.

### 4.3 EFFECT OF SUPER RESOLUTION

The primary reason for using a super-resolution model like Real-ESRGAN instead of a simple downsampling and upsampling operation is that GAN-based models aim to map the image distribution closer to that of natural images. In contrast, naive upsampling and downsampling merely perform basic averaging, which can be exploited by attackers. Specifically, an attacker can craft sufficiently smooth noise, so it remains unchanged after downsampling and upsampling. We demonstrate this in Figure 6. In this comparison, the threshold and setup remain consistent, with the only difference being that the naive

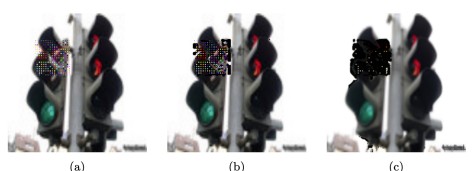

(a)      (b)      (c)

Figure 6: (a) White-box attack with the smoothed adversarial patch. (b) Results after applying the naive defense, which fails because the adversarial patch is smoothed. (c) Output of our proposed model, successfully masking the adversarial patch.

method replaces the GAN-based model with simple upsampling. The results show that, unlike our approach, the naive method fails to mask the adversarial patch effectively.

## 5 CONCLUSIONS

In this paper, we proposed a new model-agnostic defense method against both localized and distributed patch attacks. *SuperPure* utilizes discrepancies between the outputs of a reconstructed image using a super-resolution GAN and the original input to mask adversarial regions. Through extensive experiments, we demonstrated the superior robustness of our method compared to prior work, showcasing its ability to defend effectively against a variety of patch-based adversarial attacks.

---

[1]We report results for two patch sizes on a single model due to the high computational cost of white-box experiments. However, based on prior findings, we believe these results are representative and likely transferable to other models and patch sizes.

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

# A APPENDIX

## A.1 THE USE OF LARGE LANGUAGE MODEL

We employed large language model (GPT-5) to assist in proofreading the manuscript. Specifically, the LLM was used for grammar checking and minor language polishing. The scientific content, experimental design, and conclusions were solely developed and verified by the authors.

## A.2 ETHICS STATEMENT

This work uses only publicly available datasets and does not involve human subjects or private information. We follow all licensing terms of the datasets used and ensure that our methods are evaluated in a controlled research setting. Our contributions focus on methodological improvements and do not pose foreseeable risks of misuse beyond standard machine learning research.

## A.3 COMPUTATION OVERHEAD OPTIMIZATIONS

While our main goal is to improve the system's robustness against singular (localized) and distributed patch attacks, we've made a series of design decisions in order to reduce *SuperPure*'s computation overhead. In this part, we briefly discuss them and explain other alternatives.

**Super-Resolution Model.** An important component in our design is the super-resolution model. Among various options, we opt for a GAN-based model since we found that it can achieve the right balance between robustness and complexity. The alternative option is using a more sophisticated model, e.g., a diffusion-based system (Nie et al., 2022a). While we expect that such a model would perform better (in terms of robustness and accuracy), it incurs orders of magnitude higher overhead (latency, memory, etc.).

Alternatively, the other extreme is using a much simpler upsampling strategy to further reduce the complexity. While we considered this, our initial analysis showed that simpler models are significantly more vulnerable to low-noise distributed attacks. Even worse, they are far more vulnerable to adaptive white-box attacks where an adversary creates patches that are resistant to up/downsampling.

**Stop Condition.** Another important factor in optimizing end-to-end latency is the stop condition. There is a tradeoff between the number of iterations and end-to-end latency. On one hand, more iterations are necessary to enhance robustness (see Figure 2). On the other hand, fewer iterations result in lower latency. We address this balance by implementing a dynamic stop condition method (see lines 6-7 in Algorithm 1) based on a user-defined parameter ($\epsilon$). Overall, our results demonstrate that *SuperPure* can achieve high robustness without requiring an excessively large number of iterations (fewer than ten on average), enabling it to attain both efficiency and robustness simultaneously.

## A.4 WHY *SuperPure* IS EFFECTIVE AND ROBUST?

Based on our theoretical insight and empirical observations, we argue that *SuperPure* is effective and robust against a wide range of attacks due to its ability to disrupt adversarial regions while preserving benign content disproportionately. *SuperPure* leverages a guided-super-resolution model. The model struggles to reconstruct adversarial patch regions more than natural, non-patch areas.

This is due to two **key factors**: (1) adversarial regions lack the structural coherence and redundancy found in natural image content, providing less recoverable information after downsampling; and (2) GANs are trained to map inputs to the natural image distribution, causing them to suppress or distort unnatural features such as adversarial noise.

In Figure 7, we illustrate an example image where the adversarial patch region shows distinctly higher reconstruction error than non-patch areas after GAN upsampling. As a result, *SuperPure* introduces stronger, localized degradation in patch regions, enabling the subsequent masking step to more effectively remove remaining traces. This two-stage design explains why *SuperPure* performs well even under challenging adaptive attacks.

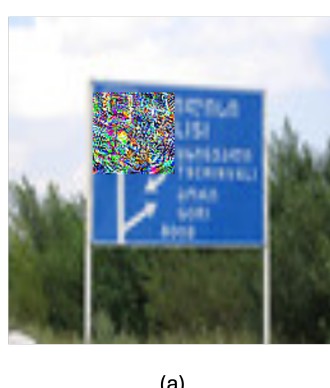 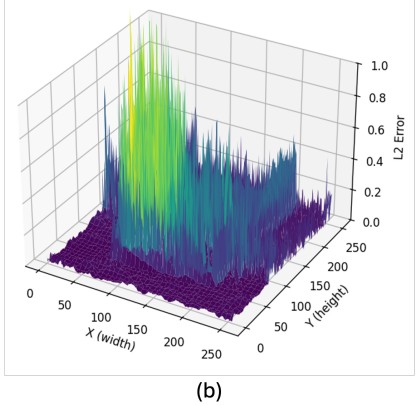

(a)                          (b)

Figure 7: (a) Example image with an applied adversarial patch disrupting a localized region, later used for reconstruction error analysis. (b) Reconstruction error across adversarial patch regions versus non-patch regions, illustrating that patched pixels exhibit significantly higher mean squared error (MSE) post-GAN upsampling.

---

**Algorithm 1:** Forward Method for *SuperPure*

---

**Input:** Input image $\mathbf{x}_{\mathrm{adv}}$, max iteration count $\mathcal{K}$, masking threshold $\lambda$, stopping criterion $\epsilon$,
enhance flag $\mathcal{E}$, downsampling function `DOWN`, pretrained superresolution model $G$

**Output:** Processed image $\mathbf{x}$

1 **Procedure** `SuperPure` $(\mathbf{x}_{\mathrm{adv}}, \mathcal{K}, \lambda, \epsilon, \mathcal{E})$:

2      **for** $k \leftarrow 0$ **to** $\mathcal{K}$ **do**

3          $\mathbf{x}_{\mathrm{down}} \leftarrow$ `DOWN` $(\mathbf{x}_{\mathrm{adv}}, 4)$;

4          $\mathbf{x}_{\mathrm{up}} \leftarrow G(\mathbf{x}_{\mathrm{down}}, 4)$;

5          $\mathbf{x}_{\mathrm{adv}}, c \leftarrow$ `GetDiff` $(\mathbf{x}_{\mathrm{adv}}, \mathbf{x}_{up}, \lambda)$;

6          **if** $c < \epsilon$ **then**

7              **break**;

8          **end**

9      **end**

10      $\mathbf{x} \leftarrow \mathbf{x}_{\mathrm{adv}}$;

11      **if** $\mathcal{E}$ **then**

12          $\mathbf{x} \leftarrow G(\mathbf{x}, 2)$;

13          $\mathbf{x} \leftarrow$ `DOWN` $(\mathbf{x}, 2)$;

14      **end**

15      **return** $\mathbf{x}$;

16

17 **Procedure** `GetDiff` $(\mathbf{x}_{\mathrm{adv}}, \mathbf{x}_{up}, \lambda)$:

18      $\mathbf{d} \leftarrow \mathcal{L}_2(\mathbf{x}_{\mathrm{adv}}, \mathbf{x}_{\mathrm{up}})$;

19      $\mathbf{m} \leftarrow \mathbf{d} > \lambda$;

20      $\mathbf{x}_{\mathrm{adv}} \leftarrow \mathbf{x}_{\mathrm{adv}} \odot \mathbf{m}$;

21      $\mathbf{c} \leftarrow \sum_{i,j} \mathbf{m}(i,j)$;

22      **return** $\mathbf{x}_{\mathrm{adv}}, \mathbf{c}$;

---

In essence, *SuperPure* does not chase evolving patch patterns—it targets what makes adversarial patches ***fundamentally*** different from natural images. This design principle ensures forward compatibility and robustness against a wide class of future, adaptive, or unseen patch-based attacks.

## A.5   ALGORITHM

The details of our purification algorithm are shown in Algorithm 1.

## A.6 Iterations to Convergence

Figure 8 demonstrates the relationship between patch size, Top-1 accuracy, and the average number of iterations to convergence for the ResNet model in the context of *SuperPure*.

As seen in the Figure, the average number of iterations needed for convergence increases as the patch size grows, suggesting that our method is able to **adapt to different patch sizes** for computational efficiency. In scenarios where there is no patch present, *SuperPure* only requires around 3 iterations on average before stopping. As the patch size increases, the number of iterations rises accordingly, reflecting the greater complexity of defending against larger adversarial patches. Note that the number of iterations for EfficientNet and ResNet is slightly different because patches are generated specifically for each classifier.

## A.7 Effect of Enhancement

An integral component of our algorithm includes the option to enable or disable a feature we refer to as "enhancement." As described in Section 3.4, this feature involves a two-step process where the input image is first up-sampled to a higher resolution and then down-sampled back to its original dimensions. In Table 2, we observe that enhancement is not only essential for defending against smaller, distributed patches but also beneficial for robustness for clean images and singular adversarial patches.

To better understand the role of enhancement, we conduct experiments where we apply this process, without iterative masking, to clean images. In addition to the ResNet and EfficientNet architectures, we evaluate three other classifiers: VGG-16 with batch normalization (Simonyan & Zisserman, 2015), WideResNet-50-2 (Zagoruyko & Komodakis, 2016), and ViT-B/16 (Dosovitskiy et al., 2021). The results reported in Table 4 reveal that top-1 accuracy increases by an average of approximately 10 percentage points. We can see in Figure 9 that enhancement improves the visual quality of images, with clearer boundaries and more defined textures, which may aid the model in focusing on key features for classification.

Table 4: The impact of enhancement on top-1 clean accuracy.

| Model | Standard | Enhanced | Change |
|---|---|---|---|
| EfficientNet-B0 (Tan & Le, 2019) | 60.76% | 70.60% | +9.84% |
| WideResNet-50-2 (Zagoruyko & Komodakis, 2016) | 61.00% | 74.46% | +13.46% |
| VGG-16 with BN (Simonyan & Zisserman, 2015) | 47.12% | 58.28% | +11.16% |
| ViT-B/16 (Dosovitskiy et al., 2021) | 74.84% | 84.82% | +9.98% |
| ResNet-152 V2 (He et al., 2016b) | 71.70% | 81.52% | +9.82% |

## A.8 Effect of Masking Threshold

Figure 10 illustrate the impact of the masking threshold, $\lambda$, on classifier accuracy and the average number of iterations until convergence. We observe that a low masking threshold leads to suboptimal

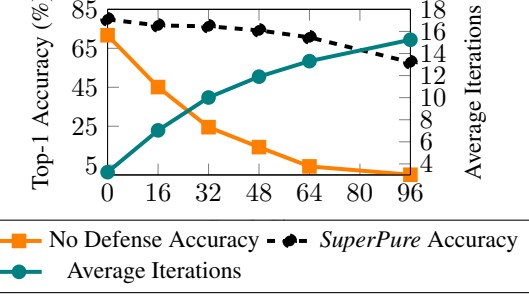

Figure 8: Relationship between patch size, accuracy, and average number of iterations until convergence.

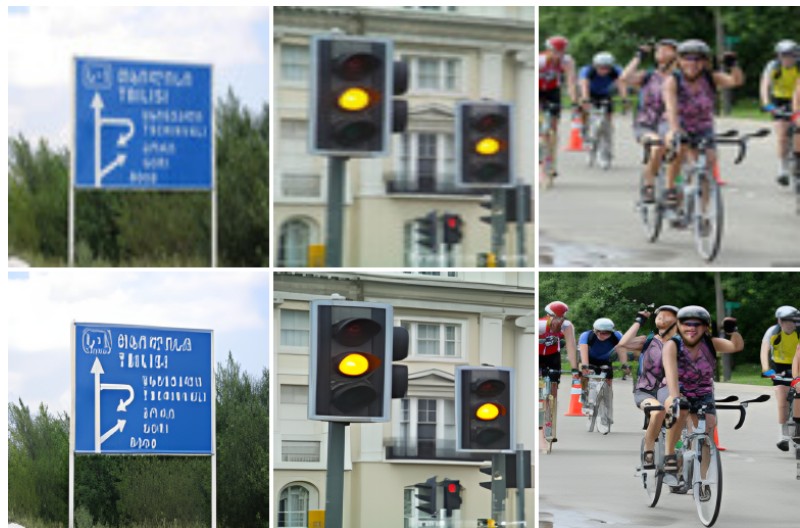

Figure 9: Clean images before (top) and after (bottom) enhancement.

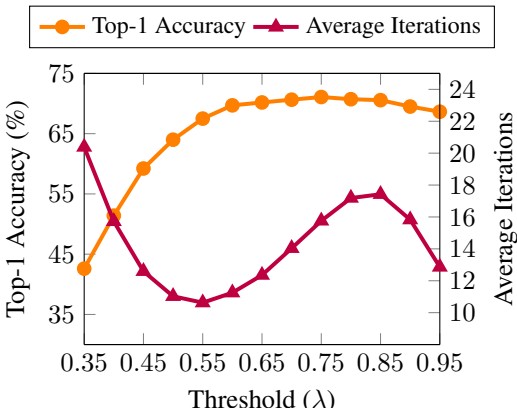

Figure 10: Effect of changing threshold ($\lambda$) on top-1 accuracy and average iterations for ResNet and Patch Size= 64.

accuracy, with the most effective range being around 0.75, although performance begins to plateau at approximately 0.6. The average number of iterations is notably higher at lower thresholds, as a lower threshold increases the likelihood of masking more pixels at each iteration. The lowest number of iterations occurs near a threshold of 0.55, but as the threshold increases beyond this point, the number of iterations rises, possibly because fewer pixels are masked per iteration. If the threshold is too high, the number of iterations drops, but accuracy also shows a slight decline.

## A.9 EVALUATION ON CIFAR-10 & CIFAR-100

We additionally tested *SuperPure* on the **CIFAR-10** and **CIFAR-100** datasets (Krizhevsky & Hinton, 2009) to assess its generalization to smaller-resolution images. We employed a ResNet-18 (He et al., 2016a) model initially pretrained on ImageNet (Deng et al., 2009) and then fine-tuned for each CIFAR dataset. Since we use a $32 \times 32$ adversarial patch, we *upscaled* each $32 \times 32$ CIFAR image to $256 \times 256$ so that the patch would not occupy the entire image, thus creating a realistic test scenario for our iterative defense.

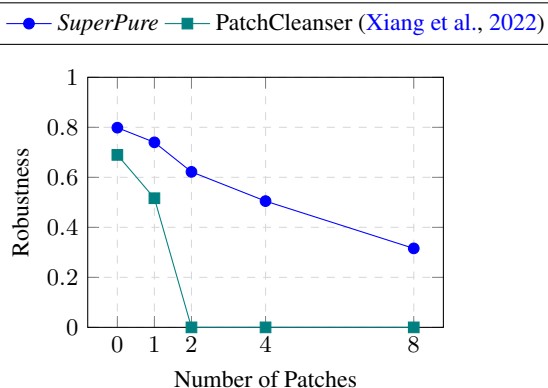

Figure 11: Comparison of the impact of increasing the number of patches in a white-box attack between our method and Patch Cleanser. Each patch measures 32×32, with a (low) noise level of 8/255.

### A.9.1 EXPERIMENTAL SETUP AND PRELIMINARY RESULTS

Table 5 shows the accuracy on clean CIFAR images, the accuracy under the $32 \times 32$ patch attack, and the recovered accuracy after applying *SuperPure*. Despite the aggressive upscaling, *SuperPure* substantially mitigates adversarial damage, suggesting that its iterative down-up masking pipeline is *dataset-agnostic* and does not rely on large native resolutions.

Table 5: CIFAR-10 and CIFAR-100 results using a fine-tuned ResNet-18 (images upscaled to $256 \times 256$).

| Dataset | Clean | Attack | After *SuperPure* |
|---------|-------|--------|-------------------|
| CIFAR-10 | 94.60% | 2.38% | 85.78% |
| CIFAR-100 | 80.09% | 2.06% | 62.26% |

### A.10 ALTERNATIVE SR: DIFFUSION-BASED MODELS

In the main paper, we selected Real-ESRGAN for super-resolution due to its relatively fast inference, which is critical for our *iterative* down-up cycles. However, recent diffusion-based SR approaches, such as **SR3** (Ho et al., 2021), can sometimes yield higher-quality reconstructions. We briefly experimented with SR3 to explore this trade-off.

Our tests show that while SR3 can improve image fidelity slightly, it is *significantly slower*. On a single image, SR3 may take *several seconds*, making multiple passes impractical. By contrast, Real-ESRGAN performs sufficiently fast to allow repeated down-up cycles. Moreover, we observed *only minor differences* in overall adversarial robustness between SR3 and Real-ESRGAN, reaffirming that *any sufficiently non-linear SR* method can disrupt patch artifacts effectively, as long as it shifts images closer to the *natural* manifold.

### A.11 DISTRIBUTED WHITE-BOX PATCHES VS. PATCHCLEANSER

We consider *distributed* adversarial patches where multiple $32 \times 32$ regions are placed throughout the image. Crucially, the attacker has **white-box** knowledge of our *SuperPure* method, specifically crafting these patches to exploit *SuperPure*'s iterative masking. We also evaluate PatchCleanser (Xiang et al., 2022) under the same multi-patch distribution for comparison, even though PatchCleanser itself does not operate in a white-box mode.

### A.11.1 EXPERIMENTAL SETUP

For each experiment, we increment the number of $32 \times 32$ patches scattered across the image. The total adversarial area thus becomes increasingly fragmented, posing a stronger challenge. While *SuperPure* faces a white-box attacker, PatchCleanser is tested as-is.

### A.11.2 Results and Observations

Figure 11 plots the robust accuracy as the number of distributed patches increases. Despite the attacker's full knowledge of *SuperPure*, our method preserves high robustness. In contrast, Patch-Cleanser (Xiang et al., 2022) degrades rapidly as more patches are introduced.

These experiments confirm that even when an attacker crafts multiple patches with internal knowledge of *SuperPure*, its repeated non-linear reconstructions continue to suppress patch effectiveness. While PatchCleanser performs well against a small number of large patches, it struggles to remain robust against numerous distributed patches. Consequently, *SuperPure* surpasses PatchCleanser in complex adversarial scenarios involving high fragmentation.

Table 6: Retraining and external–data requirements of each defense. "Additional Data" indicates the use of a pretrained model trained on an external dataset. "Retraining" refers to defenses that require retraining the classifier to function.

| Defense Method | Retraining | Additional Data |
|---|---|---|
| No Defense | No | No |
| PatchCleanser (Xiang et al., 2022) | No | No |
| PAD (Jing et al., 2024) | No | Yes |
| PatchCURE (Xiang et al., 2024) | Yes | No |
| *SuperPure* (ours) | No | Yes |

### A.11.3 Deployment Overhead

Table 6 summarizes the integration cost of each defense. PatchCleanser (Xiang et al., 2022) is a preprocessing-only approach and therefore requires *neither* classifier retraining nor extra data. PAD relies on the Segment Anything Model (SAM) (Kirillov et al., 2023) for patch localization; SAM is a foundation model trained on external data, but PAD itself leaves the downstream classifier untouched (Jing et al., 2024). PatchCURE (Xiang et al., 2024) alters the backbone (e.g. ViT-SRF) and must be partially retrained, increasing deployment complexity. Our method, *SuperPure*, similarly uses a publicly available super-resolution backbone trained outside ImageNet, yet remains fully *plug-and-play*: no classifier retraining is needed, and the added data requirement is rewarded with an 8% improvement in clean accuracy over the baseline (Table 2). Thus, *SuperPure* provides the most favorable trade-off between robustness, latency, accuracy, and engineering effort.

