# OpenReview forum: "SuperPure: Efficient Purification of Localized and Distributed Adversarial Patches"
_ICLR.cc/2026/Conference — ICLR 2026 Conference Desk Rejected Submission_

### Official Review · Reviewer_qzXk · 2025-10-28

**Soundness:** 2
**Presentation:** 3
**Contribution:** 2
**Rating:** 4
**Confidence:** 5

**Summary:**

This submission introduces SuperPure, a plug-and-play defense for vision-based ML models vulnerable to adversarial patch attacks. The method iteratively combines pixel-wise adversarial masking with GAN-based super-resolution to suppress both localized and distributed patches without requiring classifier retraining. The paper conducts extensive evaluations on the ImageNet using various architectures, attack types, and patch sizes, showing improvements in adversarial robustness, clean performance, and computational efficiency compared to selected defenses.

**Strengths:**

1. The performance improvement is impressive compared to selected adversarial patch defense baselines, demonstrating high versatility, effectiveness, and efficiency.

2. The evaluation is thorough, covering multiple model backbones, different datasets, varying patch sizes, and both white-box and several black-box attack scenarios.

3. The paper provides clear algorithmic descriptions, well figure illustrations, and reader-friendly writing, making the technical content easy to follow and reproduce.

**Weaknesses:**

1. The idea of using downsampling and upsampling through a generative model followed by threshold-based filtering of adversarial pixels has been explored previously in DiffPAD [1]. However, this prior work is not discussed or cited as the related work.

2. The paper provides limited theoretical justification for why the proposed residual-masking mechanism should be fundamentally more robust than existing certified defenses or denoising-based baselines.

3. To better position SuperPure with respect to recent adversarial purification works, the authors are expected to include the empirical comparison with diffusion-based patch removal methods, e.g. DIFFender [2], which can recover the image without the patch vestige.

4. The component-level ablation study is missing. Showing results for SuperPure 'without enhancement' or 'without iterations' would help isolate how much individual step contributes to the final performance gains.

[1] *DiffPAD: Denoising Diffusion-based Adversarial Patch Decontamination.* in WACV 2025.

[2] *DIFFender: Diffusion-based Adversarial Defense against Patch Attacks.* in ECCV 2024.

**Questions:**

1. Logically, for the mask $\mathbf{m}$, shouldn’t pixels with a difference larger than the threshold $\lambda$ be masked out, i. e., $\mathbf{m}(i,j)=0$ instead of $1$ as written around line 202?

2. Does the optimal threshold $\lambda$ vary across datasets? In the paper, $\lambda=0.7$ is used, but Fig. 10 suggests that $\lambda=0.6$ yields a better accuracy–latency trade-off.

3. What adaptive attack setting was used—BPDA, EOT, or another variant? How were the attack iterations and other key hyperparameters configured? Please provide details for reproducibility.

Please respond to both the questions and the weaknesses during the rebuttal phase.

---

### Official Review · Reviewer_FtCR · 2025-10-29

**Soundness:** 3
**Presentation:** 2
**Contribution:** 2
**Rating:** 2
**Confidence:** 4

**Summary:**

The paper introduces SuperPure, an efficient defense strategy against both localized and distributed adversarial patch attacks for vision systems. Unlike prior certified or heuristic defenses that are vulnerable to distributed patch attacks or suffer from high computational overhead, SuperPure combines iterative pixel-wise masking—driven by differences between the input and a GAN-based super-resolved image—with a final enhancement stage to efficiently remove both concentrated and dispersed adversarial perturbations. Extensive experiments are conducted across ImageNet with several architectures (ResNet, ViT, EfficientNet), demonstrating marked improvements in robustness, clean accuracy, and latency versus state-of-the-art baselines.

**Strengths:**

1. Robust empirical performance: across ImageNet and multiple attacks (localized, distributed, natural), SuperPure achieves superior robust accuracy (Table 1 and Table 2), clean accuracy (especially notable in Table 2 and Table 4, also confirmed by the clean image enhancement experiments), and drastically reduced latency (as evidenced by the “Lat” column in Table 1).

2. The analysis is sharp regarding adaptive white-box attacks;  Table 3 indicates SuperPure’s resilience in white-box settings—outperforming both naive down/up and standard PatchCleanser.

**Weaknesses:**

1.  Insufficient Ablation Studies: The manuscript lacks ablation studies on the upsampling and downsampling scale factors $s$. Furthermore, there is no discussion of ablation studies for step one (lines 2 to 9 of Algorithm 1) and step two (lines 11 to 15 of Algorithm 1), making it difficult to intuitively observe the contribution of steps one and two to the overall defense effectiveness. In addition, the hyperparameter $\epsilon$ has not been subjected to ablation studies. Why is $\epsilon=4$?

2.  Enhance Flag $\mathcal{E}$: The manuscript does not discuss the value of the enhance flag $\mathcal{E}$ in Algorithm 1. How should the enhance flag be set for different types of attacks?

3.  External Data: SuperPure utilizes a Real-ESRGAN model pretrained on non-ImageNet data, which (Table 6 in the Appendix) introduces a requirement for "external data." Although this impact is minimal in practice, it deviates from the principle of "pure plug-and-play, no external data required" advocated by some competing methods (such as PatchCleanser). A deeper exploration is needed regarding the choice of the super-resolution model and its influence on cross-dataset generalization.

4.  More Super-Resolution Architectures: The paper suggests that GAN-based upsampling is crucial and provides support, but the analysis does not delve deeper into the diversity of different super-resolution architectures.

5.  Figure 1: The SuperPure flowchart needs to be redesigned and redrawn.

**Questions:**

1. Can the authors clarify or provide empirical/statistical analysis on the robustness of the chosen threshold $\lambda$ and stopping condition $\epsilon$ across different attack strengths, benign/noisy image scenarios, and GAN variants? Is there a risk of over-masking benign pixels, or missing novel adaptive perturbations?

2. Would the authors be able to provide further ablation between different SR architectures (e.g., more advanced diffusion models, different GAN backbones) and their tradeoffs in robustness, speed, and false positive rates?

3. For Table 1, could more baselines (e.g., compression, diffusion, or randomization-based techniques) be provided to establish a more comprehensive empirical benchmark?

---

### Official Review · Reviewer_JTYd · 2025-10-31

**Soundness:** 2
**Presentation:** 3
**Contribution:** 1
**Rating:** 2
**Confidence:** 5

**Summary:**

This paper proposes SuperPure, a defense against adversarial patch attacks that combines iterative downsampling/upsampling with GAN-based super-resolution and pixel-wise masking. The authors evaluate against localized, distributed, and natural patch attacks, showing improvements over PatchCleanser and PAD.

**Strengths:**

The adversarial attacks is an unsolved problem.
The paper suggests a plug-and-play defense, which has lower time overheads than some of state-of-the-art approaches, which is commendable

**Weaknesses:**

# Major Weaknesses

## 1. Weak Theoretical Foundation and Unsupported Assumptions:

- The theoretical justification in Section 3 relies on several claims that are either unsupported or inappropriately referenced:
Inappropriate Shannon citation (line 147): Shannon (1949) addresses sampling theory for signals, not adversarial perturbations. The connection between sampling theorem and adversarial patch effectiveness is not established. This citation appears to lend unwarranted authority to claims about patches requiring "high-frequency information."

- "High-frequency" assumption lacks support (lines 145-147): The paper asserts that adversarial patches are "heavily reliant on precise, high-frequency perturbations" as if this was obvious. While Brown et al. (2017) is cited, this claim is problematic because: Naturalistic patches are explicitly designed NOT to be high-frequency noise. Distributed patches with low perturbation density (0.1% in DorPatch) may not follow this pattern

- Vague claims on natural image properties (lines 152-153): The claim that "Important features and structures are often replicated across the image, making essential information less susceptible to significant degradation" is ill-defined: What constitutes "important features"?
Does "replicated across the image" mean spatial redundancy, local correlation, or multi-scale structure? This needs precise definition with references to image statistics literature (e.g., natural scene statistics, texture synthesis), and how does this property actually translate to robustness after downsampling?


## 2. Imprecise Mathematical FormulationProblem formulation and equations are sloppy:

- Equation 1 (line 158): Writing $x_adv = x + \delta$ is misleading because patches don't add to the entire image—they occupy specific spatial regions.

- Equation 2 reasoning incomplete (lines 160-162):  The claim that $\|D_s(\delta)\|_2^2 \approx \frac{1}{s^2}\|\delta\|_2^2$ indicates ``substantial decrease in perturbation's magnitude'' misses the key point. The same scaling applies to clean content: $\|D_s(\mathbf{x})\|_2^2 \approx \frac{1}{s^2}\|\mathbf{x}\|_2^2$. What matters is the \emph{relative} degradation---the ratio $\|D_s(\delta)\|_2^2/\|D_s(\mathbf{x})\|_2^2$ compared to $\|\delta\|_2^2/\|\mathbf{x}\|_2^2$. Why should patches degrade more than natural content? The math presented doesn't establish this.

- Equation 3 imprecise (line 194): The expectation notation $\mathbb{E}(|p_{i,j}^a - G(D_s(\pi))_{i,j}|)$ lacks clarity. Expectation over what distribution? Clean images $\pi$? Patch locations? Patch content? Both? This needs to be formalized properly.

## 3. Unclear Role of Super-Resolution vs Generic GAN Projection

- Why super-resolution specifically? The paragraph starting line 184 ("There are two reasons why the GAN struggles to reconstruct patch regions...") describes properties that would apply to any GAN trained on natural images:
Reason 1: Patches lack local coherence  $\rightarrow$  this applies to any GAN trained on natural images; would struggle
Reason 2: GANs map to natural image distribution $\rightarrow$ this applies to any natural-image GAN; would do this

This raises critical questions:

- **Is super-resolution essential, or is this really about GAN-based projection to the natural manifold?**

- **What specific property of the super-resolution task makes it suitable?**

The paper needs to either: (a) provide ablations comparing different GAN types, or (b) clarify that the method is fundamentally about manifold projection and super-resolution is just a convenient, pre-trained model. The current framing suggests super-resolution is essential when the reasoning suggests otherwise.

## 4. Insufficient Adaptive Attack Evaluation
While Section 4.2.3 includes white-box attacks, the evaluation has significant limitations:
Limited scope: Only two patch sizes (48×48, 64×64) on one model (ResNet) due to "computational cost." For a defense paper, this is insufficient—adaptive attacks are the primary threat model.

Missing adaptive strategies like for example:

a. Patches optimized specifically to survive the Real-ESRGAN reconstruction
b. Smooth, low-frequency patches designed to evade the "high-frequency destruction" mechanism
c. Adaptive distributed patches aware of the iterative masking threshold
d Attacks that exploit the deterministic nature of the defense


- **Incomplete naturalistic patch evaluation:** The high-frequency assumption contradicts the paper's own results on TnT attacks (natural flower patches). If patches don't need to be high-frequency noise, why does the defense work? This suggests the mechanism is different than claimed.

- **No analysis of failure modes:** When does SuperPure fail? What patch characteristics make it vulnerable? This understanding is crucial for assessing real-world robustness.

## 5. Experimental Design and Comparison Fairness Issues


Real-ESRGAN data leakage concern (line 295): While DIV2K, Flickr2K, OST formally exclude ImageNet, what about semantic overlap? Are there scene types, object categories, or texture patterns present in both? This matters because the GAN may have implicit knowledge of ImageNet-like distributions, giving it an unfair advantage.


# Minor
- Stopping condition (Section 3.3): Why $\varepsilon=4$ pixels specifically? Is this tuned per dataset/model ?

**Questions:**

I suggest the paper needs:
- convincing insights and
- strong adaptive attacks-- e.g. :

1. Optimizing patches explicitly to minimize reconstruction error under Real-ESRGAN?
2. Smooth patches with controlled frequency content?
3. Distributed patches optimized against the full SuperPure pipeline?

---

### Official Review · Reviewer_JAiN · 2025-11-01

**Soundness:** 1
**Presentation:** 2
**Contribution:** 2
**Rating:** 2
**Confidence:** 4

**Summary:**

The paper proposes SuperPure, a plug-and-play defense that iteratively downsamples an input, upscales it with a GAN super-resolution model (Real-ESRGAN), and masks pixels with large reconstruction residuals (between the original input and the GAN output). A final enhancement step (GAN upsampling then downsampling) targets low-amplitude distributed noise. Evaluations on ImageNet and three backbones (ResNet-152, ViT-B/16, EfficientNet-B0) report robustness gain against localized and distributed patches, with lower latency than PatchCleanser/PatchCURE/PAD on Jetson Orin Nano.

**Strengths:**

- Practicality & speed. SuperPure is model-agnostic and significantly faster than prior defenses on edge hardware; the paper reports 0.72s/img vs >50s for PatchCleanser and 12s for PatchCURE on Jetson Orin Nano.
- Coverage of threat models. Results span localized, distributed (DorPatch), and naturalistic (TnT) patches, with strong distributed-attack robustness where certified defenses collapse.
- Clean accuracy improvements (∼+8–10% across backbones) attributed to the enhancement step are intriguing and empirically supported.
- White-box adaptive attempts. SuperPure shows >60%/51% robust accuracy for 48×48/64×64 patches on ResNet.

**Weaknesses:**

### 1. Theoretical claim around Eq. (2) and line 162 is overstated without conditions.
The paper derives Eq. (2) and concludes this indicates “a substantial decrease” in the perturbation magnitude due to spatial averaging.  The same downsampling operator $D_s$ applies to the clean image x: $\|D_s(x)\|_2^2 \approx \frac{1}{s^2}\|x\|_2^2$. Without frequency-content assumptions, the relative proportion $\frac{\|D_s(\delta)\|_2}{\|D_s(x)\|_2}$ is unchanged (the common factor cancels). The claim that downsampling disproportionately suppresses $\delta$ requires explicit conditions (e.g., $\delta$ has materially higher energy above the anti-alias cutoff than x), not just Eq. (2). The current text cites general “high-frequency” intuition but doesn’t quantify the spectral separation or aliasing kernel properties for bilinear downsampling.

### 2. Theoretical claim around Eq. (3) lacks clearly stated conditions and can fail for GAN-generated patches (TnT, IAP[a]).
Equation (3) asserts that adversarial pixels yield much larger reconstruction error than clean pixels after G(D_s(\cdot)). This depends critically on a distribution gap between patch textures and the natural manifold learned by the same class of GANs. For GAN-generated patches (e.g., IAP) that explicitly optimize natural-looking textures, or for patches produced by a generator similar to the defense’s SR model, residuals can be small—contradicting Eq. (3)’s premise at the threshold level $\lambda$. The paper does evaluate TnT (natural patches) and indeed SuperPure trails PAD there (43.2% vs 48%).

[a] Inconspicuous adversarial patches for fooling image-recognition systems on mobile devices

### 3. Adaptive-attack section is under-specified; unclear if attacks are sufficiently strong.
The paper claims full white-box adaptivity (gradients through SR and the pipeline) and reports strong robustness (Table 3), but critical details are missing: loss functions, step sizes/steps, restarts, EOT over any stochasticity, and (crucially) how the non-differentiable thresholded mask is handled (e.g., smoothed surrogate, STE). Without these, it’s hard to judge whether the reported white-box numbers represent a worst-case attacker.

### 4. Enhancement-driven clean-accuracy gains are likely distribution-coupled
The main experiments are on a 5k-image subset of ImageNet across three backbones. The defense uses Real-ESRGAN for SR and explicitly states “lack of ImageNet training overlap” (models trained on DIV2K/Flickr2K/OST), then fixes default hyper-params. A key claim is sizable clean-accuracy gains from the enhancement step). The appendix adds CIFAR-10/100 (upscaled to 256×256 with a fine-tuned ResNet-18) and asserts the pipeline is “dataset-agnostic,” based on those results.
Though Real-ESRGAN wasn’t trained on ImageNet per se, it is trained on natural photographic imagery very similar to ImageNet’s distribution. The enhancement module (upsample→downsample) can therefore “polish” ImageNet-like textures and edges in ways that classifiers reward on ImageNet, inflating clean accuracy in-distribution. This effect can evaporate—or even become harmful—under distribution shift (e.g., sketches, renderings, domains with different texture statistics), meaning the “dual benefit” claim (robustness and clean-accuracy ↑) may not hold generally. The current paper does not test on OOD benchmarks where SR priors are mismatched to the downstream data—so we cannot tell whether the gains persist beyond ImageNet-like photos. (The CIFAR check is limited: heavy upscaling with a small-image model and a single backbone; it does not probe natural-image distribution shift at scale.)

### 5. Methodological novelty
The paper’s pipeline—downsample → generative upsample (SR) → residual-threshold mask with iterative replacement → final up–down “enhancement”—is not fundamentally distinct from well-studied generative/purification and frequency-suppressing patch defenses. It primarily recombines known primitives (frequency attenuation + projection to a natural-image prior + localized overwrite) that have appeared separately in the literature. The manuscript should more carefully position its contribution and provide method-centric evidence of novelty.

**Questions:**

1. Precisely state the assumptions under which $\|D_s(\delta)\|_2^2 \approx \| \delta \|_2^2/s^2$ yields a relative attenuation vs. $\|D_s(x)\|_2^2$. E.g., are you assuming the patch’s spectrum has more mass above the anti-alias cutoff than the clean image? If yes, quantify this. Specify the downsampling kernel (bilinear interpolation) and its effective low-pass characteristics used in your derivation. How do aliasing and prefiltering enter into Eq. (2)?
2. State explicit preconditions for Eq. (3): e.g., manifold mismatch between patch textures and the SR prior, and/or context incoherence after downsampling. Is Eq. (3) intended as a stochastic inequality (“on average over samples”)? If so, define the distribution(s).
3. For adaptive white-box setup, provide (i) Exact objective(s) (e.g., cross-entropy on target, any auxiliary penalties). (2) Whether you use EOT over any internal randomness. (iii) Treatment of the binary mask. and important details relative to the worst-case implementation.
4. Explore clean-accuracy gains under dataset OOD settings.
5. Clarify in what precise method principle SuperPure differs from prior generative purification (SR/diffusion), frequency/gradient smoothing, and localize-then-repair defenses? Which failure modes of generative-model-based patch defenses does SuperPure specifically fix, and why (mechanistically)?

---

### Official Review · Reviewer_7o18 · 2025-11-02

**Soundness:** 2
**Presentation:** 2
**Contribution:** 1
**Rating:** 2
**Confidence:** 4

**Summary:**

This paper proposes SuperPure, a "plug-and-play" defense against adversarial patch attacks designed for efficiency, particularly on edge devices. The method aims to defend against both localized patches and low-noise distributed patches (like DorPatch). The core mechanism is a two-part process: (1) an "Iterative Masking" step that downsamples the image, uses a GAN (Real-ESRGAN) for upsampling, and then masks pixels with a high reconstruction residual (difference) between the original and reconstructed image. This process repeats until convergence. (2) A final "Enhancement Step" that performs a single GAN-upsample followed by a downsample, which the authors claim removes subtle, distributed perturbations. The paper claims state-of-the-art results, showing a 10x speedup on a Jetson Orin Nano, 59% robustness against distributed attacks (where others allegedly fail), and an unexpected boost in clean accuracy.

**Strengths:**

- The paper addresses a significant and practical problem: the need for efficient, on-device defenses against adversarial patches, which is critical for applications like autonomous systems.

- The authors' focus on latency is commendable. Measuring performance on an actual edge device (Jetson Orin Nano) is a strong point and highly relevant to the paper's motivation.

**Weaknesses:**

- Expanding SOTA Comparisons: The current evaluation provides a good baseline, but the paper's claims of state-of-the-art performance would be significantly more comprehensive if compared against a wider set of recent defenses. The authors may want to consider including recent work such as Jedi (CVPR'23), DIFFender (ECCV'24), and NAPGuard (CVPR'24) to provide a more complete picture of the current landscape.

- Aligning Motivation and Evaluation Tasks: There appears to be a potential gap between the paper's strong motivation, which cites "autonomous driving scenarios," and its experimental validation. Since autonomous systems heavily rely on object detection, the paper's focus on classification-only tasks feels incomplete. Expanding the evaluation to include object detection would more directly support the paper's motivating claims and broaden its impact.

-  Exploring a Unified Framework: The current methodology relies on two distinct components: "Iterative Masking" for localized patches and a separate "Enhancement Step" for distributed patches. This suggests that the core masking technique may have limitations. The paper would be strengthened by a discussion on whether a more unified framework could address both attack types simultaneously, or by further clarifying the design choice for this two-part solution.

-  Analyzing the Robustness vs. Latency Trade-off: The justification for using GANs over diffusion models is noted as "speed," which is a valid and important practical consideration. However, this leaves an key scientific question open. A direct comparison against a diffusion-based method, perhaps presented on a "Robustness vs. Latency" plot, would provide a more complete analysis of the trade-offs involved and better situate the paper's contribution.

- Analysis of Failure Modes/Information Preservation: A deeper analysis of the method's potential failure modes would be beneficial. For instance, the iterative masking shown in Figure 2 appears to "black out" the patched region. This raises a question: what happens when a patch covers a semantically critical region (e.g., an eye, a car's headlight)? It would be insightful to explore whether the method risks "destroying" key information along with the patch, which might lead to misclassification for a different reason. Clarifying this would help in understanding the method's practical limitations.

**Questions:**

- To further contextualize the paper's strong performance, it would be highly beneficial to see a direct comparison (in terms of both robustness and latency) with other recent defenses like Jedi (CVPR'23), DIFFender (ECCV'24), and NAPGuard (CVPR'24). Could the authors comment on how SuperPure might position against these methods?

- The motivation of "autonomous driving" is very compelling. To fully realize this connection, have the authors considered evaluating SuperPure on object detection tasks? Results on standard benchmarks like COCO or KITTI would significantly strengthen the paper's practical claims for this domain.

- I was interested in the two-part methodological design. Could the authors elaborate on the synergy between the "Iterative Masking" and the "Enhancement Step"? Specifically, since the Enhancement Step is noted as effective for distributed patches, what unique role does the iterative process play that the enhancement step alone cannot fulfill? This would help clarify the necessity for the combined approach.

- Regarding the purification process, it would be insightful to understand its behavior in challenging scenarios. For example, when an adversarial patch occludes a semantically critical feature (like a person's face or text on a sign), what does the GAN-based reconstruction typically produce? Is there a risk that the purification process might remove the critical feature along with the patch (as the mask in Figure 2 might imply), potentially leading to a misclassification for a different reason?

---

### Note · Program_Chairs · 2026-01-17
**Submission Desk Rejected by Program Chairs**

The following references in this submission do not refer to real documents and/or have major errors in bibliographic information:

     Chang Mao, Yihan Zhu, Neil Zhenqiang Gong, and Xiang Zhang. Defending against adversarial attacks by randomized diversification. IEEE Transactions on Pattern Analysis and Machine Intelligence, 44(3):1313-1327, 2022.